# Knee Extensor Muscle Strength Associated with the Onset of Depression in Older Japanese Women: The Otassha Study

**DOI:** 10.3390/nu16142179

**Published:** 2024-07-09

**Authors:** Takahisa Ohta, Narumi Kojima, Yosuke Osuka, Hiroyuki Sasai

**Affiliations:** 1Research Team for Promoting Independence and Mental Health, Tokyo Metropolitan Institute for Geriatrics and Gerontology, Tokyo 173-0015, Japansasai@tmig.or.jp (H.S.); 2Integrated Research Initiative for Living Well with Dementia, Tokyo Metropolitan Institute for Geriatrics and Gerontology, Tokyo 173-0015, Japan; 3Department of Frailty Research, Center for Gerontology and Social Science, Research Institute, National Center for Geriatrics and Gerontology, Obu 474-8511, Japan

**Keywords:** geriatric depression, muscle strength, prospective cohort study, Japanese women

## Abstract

We examined whether knee extensor muscle strength can predict the onset of depressive symptoms in older Japanese women living in Tokyo. A baseline comprehensive geriatric examination was conducted to evaluate isometric knee extensor muscle strength and depressive symptoms (using Geriatric Depression Scale [GDS]) in 2017–2019. A free of neurological disease participants received a series of follow-up examinations following an initial evaluation. A GDS score of ≥5 during follow-up marked the onset of depressive symptoms. A logistic regression model was established after adjustment of baseline GDS score for variables including age, body mass index, smoking, alcohol consumption, comorbidities, working status, hobbies, volunteering, years of education, and dietary variety. Of the 1845 recruited individuals, 1409 were eligible to be targeted for follow-up. Among them, 768 women provided two-year follow-up data and contributed the final analysis. After covariate adjustments, the odds ratios (95% confidence interval) for depressive symptoms were 0.68 (0.39, 1.20) and 0.48 (0.26, 0.91) for the middle and highest tertiles of muscle strength, respectively, using the lowest tertile as reference. A dose-response association between muscle strength and depression (*p* = 0.022) was identified. This study suggests an inverse dose-response relationship between knee extensor muscle strength and the onset of depressive symptoms in older Japanese women.

## 1. Introduction

The prevalence of geriatric depression, a major psychiatric disorder, is rising globally, leading to critical illness, economic burden, dementia, suicide, and mortality in an aging society [1,2]. Geriatric depression can result in social isolation and is a disorder of serious concern, especially in Japan [3]. Socioeconomic status, relocation and bereavement, history of depression, chronic illness, and low physical function have been identified as risk factors for depression [1]. Although pharmacological approaches are widely employed in treating depression, the substantial increases in healthcare costs present a concern for the geriatric population undergoing pharmacotherapy for non-communicable diseases with high healthcare expenses [4]. Additionally, considering that antidepressant prescriptions account for less than 50% of prescriptions in late-life adults diagnosed with depression—who are at increased risk for polypharmacy—it is imperative to identify and study modifiable risk factors for depression [5].

Physical fitness has received attention worldwide as a potentially modifiable risk factor for depression. A report from the UK Biobank, which followed nearly 160,000 people for almost 10 years, found that the hazard ratio for developing depression was 1.24 times higher in the group with the lowest grip strength at baseline than in the group with the highest grip strength [6]. However, it has been suggested that a variety of muscle groups, such as lower extremity, should be assessed, rather than only grip strength, since the decline in lower-limb muscle strength, rather than grip strength, precedes the decline in physical function in the aging process [7].

Lower-extremity muscle strength reflects a person’s amount of physical activity, and a bidirectional relationship exists between lower-extremity muscle strength and moderate-to-vigorous physical activity [8]. However, no such relationship has been reported for grip strength [7].

The causal relationship between lower-extremity muscle strength and the onset of depressive symptoms has been reported, as examined through a follow-up of over seven years [9]. Although the authors revealed that low lower-extremity muscle strength is a risk factor for developing depressive symptoms, the age group of participants in their study was markedly wide-ranging, from middle-aged to older adults. Therefore, the causal relationship between age-related decline in lower-extremity muscle strength and the onset of depressive symptoms, specifically in older adults, remains unknown. In addition, to the best of our knowledge, there has been little evidence linking lower-extremity muscle strength and depression except for the one report above. Accordingly, Japan’s super-aged society, in which the percentage of the population aged 65 and over in the total population (aging rate) exceeds 21%, further necessitates additional studies involving older individuals.

In the present study, we aimed to determine whether knee extensor muscle strength can predict the onset of geriatric depression among older Japanese women over a two-year follow-up period. These results may be useful for promoting health and longevity in older individuals.

## 2. Methods

### 2.1. Study Design and Participants

This prospective cohort study involved participants from The Otassha Study, a large observational cohort study that involved participants sampled from the basic resident register who lived around the Tokyo Metropolitan Institute for Geriatrics and Gerontology (TMIG), Itabashi Ward, Tokyo, Japan. The study protocol has been described in an earlier study [10]. Participants commuted to the research institution independently, and baseline and follow-up examinations were conducted in the TMIG examination room. Baseline examinations were conducted in 2017, 2018, and 2019, and participants received follow-up appointments for two years. Japanese women over the age of 65 years were included in the present study. Exclusion criteria were as follows: (1) history of depression, (2) ongoing depression (Geriatric Depression Scale [GDS] ≥ 5 or anti-depression drug users), (3) missing physical fitness test at baseline, and (4) prior diagnosis and treatment of a major psychiatric or neurodegenerative illness such as dementia, Parkinson’s disease, Alzheimer’s disease, or stroke.

All participants provided written informed consent, and the Ethics Committee of the TMIG approved the study protocol (R21-16 and R22-048 both were approved on 11 October 2023).

### 2.2. Assessment of Knee Extensor Muscle Strength

The knee extensor muscle strength (N) was assessed using a handheld isometric dynamometer (μTas F-1, ANIMA, Chofu, Japan). Participants sat on a horizontal chair with their knees bent at a 90° angle, and the instructor conducted warm-up and practice exercises. The dynamometer sensor was placed on the front of the ankle 5 cm above the lateral malleolus. Each participant was instructed to perform maximal voluntary knee extensor muscle contraction. The test was conducted twice, and the superior result was recorded. The assessment protocol has been described previously [11].

### 2.3. The Onset of Geriatric Depressive Symptoms

The primary outcome was the onset of geriatric depressive symptoms, assessed using the GDS-15 [12]. The GDS-15 is commonly used to assess depressive symptoms in older participants [13]. This scale consists of 15 questions covering areas such as “basically satisfied with your life” and “dropped many of your activities and interests”, and required the participants to answer “Yes” or “No”. The scores ranged from 0 to 15, and a high GDS-15 score indicated poor mental health. Herein, a GDS score of ≥5 indicated the onset of depressive symptoms [14].

### 2.4. Assessment of Covariates

This study included several covariates such as age, body mass index (BMI), lifestyle disease (hypertension, heart disease, diabetes, dyslipidemia, osteoporosis, and cancer) [15], smoking [16], alcohol consumption [17], job engagement, hobbies or lessons [18], volunteering [18], living alone [19], education level [20], and dietary variety [21]. The dietary pattern score was calculated based on the frequency of consumption of soy products, fruits, and green-yellow vegetables. Participants received 1 point for rare consumption, 2 points for consuming these items once or twice a week, 3 points for consumption every other day, and 4 points for almost daily consumption. The total possible score ranged from 3 to 12, with higher scores suggesting a dietary pattern more conducive to mitigating depression. These covariates were assessed based on a self-reported questionnaire with some assistance from the observer.

### 2.5. Statistical Analysis

The participants were divided into tertiles based on their baseline muscle strength. Continuous variables are presented as mean and standard deviation (SD), and categorical variables are presented as percentages.

A logistic regression model using the onset of geriatric depressive symptoms as the outcome variable and the category of muscle strength as the exposure was established. A crude analysis, as well as an analysis fully adjusted for confounding factors including age, BMI, smoking, drinking habits, baseline GDS score, job engagement, living alone, hobbies or lessons, volunteering, dietary variety, education level, and medication for lifestyle diseases (hypertension, diabetes, dyslipidemia, osteoporosis, heart disease, and cancer), was performed. Multivariable odds ratios (ORs) and 95% confidence intervals (CIs) were calculated using the tertile with the lowest muscle strength as the reference. Additionally, to investigate the presence of a dose-response association between muscle strength and depressive symptoms, a trend test was conducted using muscle strength as a continuous variable in the multivariate logistic model. GDS is utilized to detect depressive symptoms, and its cutoffs vary. Therefore, in this study, in addition to using a cutoff score of 5, a sensitivity analysis was conducted in which a score of 6 or higher was considered indicative of depressive symptoms [22].

Statistical analyses were conducted using R version 4.2.2.

## 3. Results

From 2017 to 2019, 1845 older Japanese women participated in the baseline examination. Of these, 1409 women who met the eligibility criteria were followed up, and the 768 with no missing data were included in the analysis (Figure 1).

Table 1 summarizes the participant characteristics based on the knee extensor muscle strength tertiles. Overall, 96 (12.5%) participants developed geriatric depressive symptoms during the two-year follow-up period. Participants in the tertile with the lowest knee extensor muscle strength were older, exhibited a higher prevalence of obesity, and had a lower dietary variety when compared with those in the tertile with the highest muscle strength. Prevalence of living alone in the lowest-muscle-strength group seemed to be higher than those in the highest group. In contrast, engagement in hobbies or lessons was likely to be higher in the highest-muscle-strength group than in the lowest.

Table 2 shows the association between each covariate and geriatric depression. Old age was positively associated with geriatric depression. Individuals engaged in hobbies and lessons had a lower OR for depression than those who did not engage in these activities. In addition, compared to the group with less than 9 years of education, the OR of onset of geriatric depression was 0.53 (0.28, 1.00) in the group with more than 12 years of education.

Univariable analysis revealed an inverse relationship between knee extensor muscle strength and the onset of geriatric depression (Table 3). The OR (95% CI) for the highest-muscle-strength tertile was 0.34 (0.19, 0.59) and for middle was 0.52 (0.31, 0.86), using the lowest muscle strength as reference. The multivariable OR for the tertile with the highest muscle strength was 0.48 (0.26, 0.91) when compared with that of the tertile with the lowest muscle strength, and a dose-response association between muscle strength and depression was observed (*p* = 0.002).

Table 4 describes the results of the sensitivity analysis. A GDS score of ≥6 is indicative of geriatric depression. The results of the multivariate analysis adjusted for potential confounders revealed an OR of 0.37 (0.16, 0.86) for the tertile with the highest muscle strength, further validating a dose-response association between muscle strength and risk of depression.

## 4. Discussion

In the present study, older women in the highest knee extensor muscle strength tertile had a reduced risk for depressive symptoms (by approximately 50%) when compared with those in the lowest tertile after adjusting for potential confounding factors. Furthermore, this finding was consistent in a sensitivity analysis with a modified cutoff for the GDS. This finding suggests that high muscle strength plays a key role in preventing the development of depression later in life and may be beneficial for realizing healthful longevity.

High knee extensor muscle strength may play a role in the onset of geriatric depressive symptoms after adjusting for social factors and other covariates. Several longitudinal studies have also reported that low muscle strength is an independent risk predictor for depression among middle-aged and older adults [23,24,25,26]. Additionally, a recent longitudinal study revealed that among Asians, varying muscle strengths could be predictors for depression over a longer period [9]. The present study also supports the hypothesis that low muscle strength is an independent risk factor for depression.

Furthermore, physical performance is reported to be a good predictor of the development of depression [27]. This suggests that a comprehensive measure of physical function, such as the Short Physical Performance Battery, may be useful in clinical practice. In relation to the results of this study, a low level of physical function or a low level of muscle strength in the lower limbs is likely to be associated with functional limitations, which may lead to lower levels of physical activity [28]. Habitual and sufficient moderate-to-vigorous physical activity (MVPA) is considered one of the surrogate markers of health outcomes in older adults. Assessment of lower-extremity muscle strength, which is bidirectionally associated with MVPA, may be an essential assessment in addition to grip strength [8,29]. Although the implications of this study do not negate the application of grip strength in the comprehensive assessment of the geriatric population, the findings provide an opportunity to propose the need for active assessment of lower-extremity muscle strength in addition to grip strength assessment.

One plausible mechanism would be the involvement of oxidative stress and increased inflammatory responses associated with low muscle strength. Participants with reduced muscle strength generally exhibit increased levels of inflammatory cytokines (e.g., interleukin-6) and acute phase proteins (e.g., C-reactive protein) responsible for systemic inflammatory responses. Inflammatory responses may play an important role in the pathophysiology of mood disorders [30,31,32,33]. Skeletal muscle contraction may have a preventive effect on the development of mental disorders because skeletal muscle secretes myokines, which exert neuroprotective and other anti-inflammatory functions [34]. Another possible mechanism may be attributed to a fear of falling or fracture owing to lower muscle strength leading to reduced mobility, which, in turn, leads to decreased social interactions and social support, thereby increasing the risk of depression [35]. Therefore, the anti-inflammatory response associated with reduced muscle strength, along with the anxiety related to low muscle strength, may increase the risk of developing depressive symptoms.

Our findings may provide fundamental insight into the effectiveness of exercise as a preventive measure against the onset of depression. Conventionally, several types of exercise have been identified to substantially reduce depressive symptoms. Although aerobic exercise alone or combined with resistance exercise can markedly improve depressive symptoms, resistance exercise alone is not particularly effective [36]. Nevertheless, resistance exercise is the most efficient strategy to improve whole-body muscle strength and is beneficial in reducing the risk of falling and preventing sarcopenia and frailty [37]. Therefore, resistance exercises of appropriate intensity and frequency are a potential non-pharmacological therapeutic option for depression.

Furthermore, based on the results of the present study, there is a need for large-scale surveys that measure whole-body muscle strength and define the threshold of muscle strength that predisposes patients to depressive symptoms. The measurement of whole-body muscle strength should not be limited to grip strength, given that this study found that employing knee extensor muscle strength can also screen for individuals at high risk of developing depressive symptoms.

The strength of this study lies in its strict participant selection. This study sampled participants from the basic resident register in Itabashi Ward, Tokyo, ensuring high sample representativeness. Furthermore, the prospective cohort design of this study allowed us to investigate the preventive effects of muscle strength on the development of depression.

Nevertheless, several limitations should be addressed. First, we used GDS, a subjective scale with possible measurement bias. In clinical practice, the GDS score is often used to screen depressive symptoms and is a supportive tool for a depression diagnosis. Therefore, the incidence of geriatric depression in this study may be underestimated or overestimated. Thus, a longitudinal analysis with a definite outcome, such as a physician’s diagnosis, should be conducted in the future. Second, this study uses a snapshot of lower-extremity muscle strength, which may not reliably reflect regular daily muscle exercises. To effectively evaluate the usefulness of muscle strength, additional information regarding daily objective physical activity must be included, although this was not incorporated into our data set. Finally, of the 1409 participants targeted for follow-up, 641 (45.5%) could not be followed, for unknown reasons. The reason for the missing data cannot be determined and may have biased the results. It may also be necessary to consider including immune-mediated diseases, such as celiac disease [38], which manifests with symptoms of indigestion, in the investigation. Notably, multifaceted interventions, including a strict gluten-free diet and supplementation, have been reported to effectively alleviate depression in patients with celiac disease. However, given that the prevalence of celiac disease in Japan is approximately 0.05% [39], the results of this study are unlikely to be impacted by this disease. Further validation with a larger sample size is needed. Additionally, assessing lower-extremity muscle strength is less convenient than measuring grip strength and lacks standardization. Therefore, further accumulation of knowledge in this field is essential.

## 5. Conclusions

In conclusion, knee extensor muscle strength may play a key role in preventing the onset of geriatric depression in older Japanese women. Our study suggests that interventions aimed at enhancing muscle strength may help prevent the onset of depressive symptoms in older adults. Based on our findings, the focus should be on developing exercise programs specifically targeted at increasing muscle strength. Such programs could potentially be incorporated into routine geriatric care to mitigate the risk of depression. Additionally, this study will likely play a crucial role in further elucidating the relationship between skeletal muscle and mental health. Further research is needed to determine the relationship between muscle strength and mental health in the older population.

## Figures and Tables

**Figure 1 nutrients-16-02179-f001:**
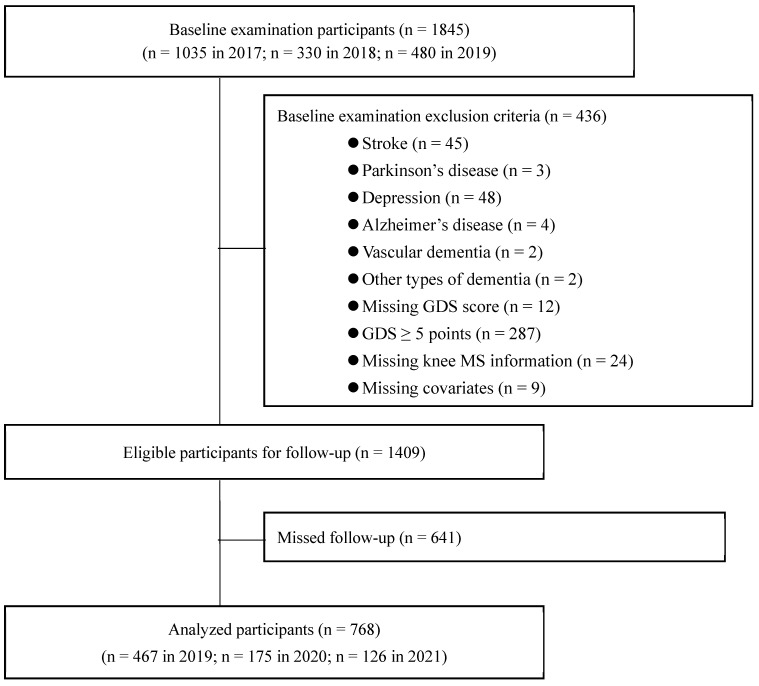
The flow of participant selection. GDS, geriatric depression scale; MS, muscle strength.

**Table 1 nutrients-16-02179-t001:** Baseline participant characteristics (2017–2019).

	Overall (n = 768)	Muscle Strength		
		Lowest (n = 256)	Middle (n = 256)	Highest (n = 256)
Number of cases	96 (12.5)	49 (16.4)	28 (9.8)	19 (7.5)
Age, years	72.7 (4.9)	73.7 (4.9)	72.9 (5.0)	71.6 (4.7)
65–69, n (%)	238 (37.1)	58 (28.5)	79 (38.3)	101 (44.4)
70–74, n (%)	257 (40.0)	88 (43.9)	86 (37.9)	83 (38.3)
75–79, n (%)	200 (20.2)	77 (23.4)	65 (21.0)	58 (16.4)
80–84, n (%)	66 (2.6)	30 (4.2)	23 (2.8)	13 (0.9)
>85, n (%)	7 (0.6)	3 (1.1)	3 (1.2)	1 (0.4)
Height, cm	152.1 (5.3)	151.9 (5.0)	151.9 (5.3)	152.7 (5.6)
Weight, kg	55.3 (8.0)	55.3 (9.0)	51.5 (7.4)	50.5 (6.6)
Body mass index, kg/m^2^ *	22.7 (3.2)	24.0 (3.6)	22.3 (2.9)	21.7 (2.7)
Lean	56 (7.8)	10 (3.7)	20 (8.4)	26 (11.2)
Normal	556 (71.5)	156 (61.2)	194 (74.3)	206 (79.0)
Obese	156 (20.7)	90 (35.0)	42 (17.3)	24 (9.8)
Knee extensor strength, N	4.31 (1.04)	3.16 (0.56)	4.32 (0.26)	5.44 (0.55)
Medication, n (%)				
Hypertension	268 (32.6)	113 (42.5)	88 (32.7)	64 (22.4)
Diabetes	78 (10.0)	37 (12.6)	24 (9.3)	17 (7.9)
Dyslipidemia	281 (34.3)	109 (39.3)	92 (33.2)	80 (30.4)
Osteoporosis	182 (21.0)	68 (23.8)	60 (19.2)	54 (20.1)
Heart disease	88 (10.0)	44 (13.6)	22 (9.8)	22 (6.5)
Cancer	99 (12.9)	37 (13.1)	28 (12.6)	34 (13.1)
Smoking, n (%)	47 (6.5)	18 (6.5)	14 (5.6)	15 (7.5)
Alcohol, n (%)	325 (43.8)	95 (41.6)	121 (43.9)	109 (45.8)
Job engagement, n (%)				
≥35 h/week	65 (65.4)	26 (66.4)	13 (68.7)	26 (61.2)
<35 h/week	185 (25.4)	52 (23.4)	64 (24.3)	69 (28.5)
None	518 (9.2)	178 (10.3)	179 (7.0)	161 (10.3)
Living alone (yes, %)	199 (22.6)	70 (26.2)	61 (22.0)	68 (19.6)
Hobby or lesson (yes, %)	483 (62.5)	148 (57.5)	165 (66.4)	170 (63.6)
Volunteering (yes, %)	296 (38.5)	91 (38.3)	114 (43.5)	91 (33.6)
Education level, years				
≤9	90 (10.1)	48 (14.5)	18 (7.5)	24 (8.4)
<9, ≤12	369 (47.7)	114 (48.1)	129 (44.9)	126 (50.0)
>12	309 (42.2)	94 (37.4)	109 (47.7)	106 (41.6)
Dietary patterns	11.11 (1.2)	10.97 (1.39)	11.13 (1.21)	11.23 (1.10)

Data are shown as the mean (standard deviation) or as numbers (percent). *: Body mass index; Lean <18.5, Normal ≤18.5, <25, Obese ≥25.

**Table 2 nutrients-16-02179-t002:** Univariate association between covariate and depressive symptoms.

Variables	Participants	Number of Cases	Incident Ratio (%)	OR (95% CIs)	*p* Value
Age, years	768	96	12.5	1.07 (1.03, 1.12)	<0.001
Body mass index, kg/m^2^					
Lean	56	5	8.9	0.67 (0.26, 1.74)	0.41
Normal	556	71	12.8	Ref	
Obese	156	20	12.8	1.01 (0.59, 1.71)	0.99
Knee extensor strength					
Lowest	256	49	19.1	Ref	
Middle	256	28	10.9	0.52 (0.31, 0.86)	0.01
Highest	256	19	7.4	0.34 (0.19, 0.59)	<0.001
Medication, yes					
Hypertension	268	40	14.9	1.39 (0.90, 2.15)	0.14
Diabetes	78	10	12.8	1.03 (0.51, 2.08)	0.93
Dyslipidemia	281	39	13.9	1.22 (0.79, 1.88)	0.38
Osteoporosis	182	26	14.3	1.23 (0.76, 1.99)	0.41
Heart disease	88	14	15.9	1.38 (0.75, 2.56)	0.31
Cancer	99	18	18.2	1.68 (0.86, 2.96)	0.07
Smoking, yes	47	9	19.1	1.73 (0.81, 3.69)	0.16
Alcohol, yes	325	39	12.0	0.92 (0.60, 1.43)	0.72
Job engagement					
≥35 h/week	65	6	9.2	1.00 (0.38, 2.64)	0.99
<35 h/week	185	17	9.2	1.61 (0.67, 3.87)	0.28
None	518	73	14.1	Ref	
Living alone, yes	199	28	14.1	1.21 (0.75, 1.94)	0.44
Hobby or lesson, yes	483	50	10.4	0.60 (0.39, 0.92)	0.02
Volunteering, yes	296	34	11.5	0.86 (0.55, 1.34)	0.50
Education level, years					
≤9	90	17	18.9	Ref	
<9, ≤12	369	45	12.2	0.60 (0.32, 1.10)	0.10
>12	309	34	11.0	0.53 (0.28, 1.00)	0.05
Dietary pattern	768	96	12.5	0.91 (0.78, 1.07)	0.27

OR, odds ratio; CI, confidence interval.

**Table 3 nutrients-16-02179-t003:** Relationship between knee extensor muscle strength and depressive symptoms (GDS ≥ 5).

	Knee Extensor Muscle Strength (N)			*p* for Trend
	Lowest	Middle	Highest	
Crude adjusted	1.00 (reference)	0.52 (0.31, 0.86)	0.34 (0.19, 0.59)	<0.001
Age adjusted *	1.00 (reference)	0.54 (0.32, 0.91)	0.37 (0.21, 0.67)	<0.001
Multivariable adjusted **	1.00 (reference)	0.68 (0.39, 1.20)	0.48 (0.26, 0.91)	0.022

Values are expressed as odds ratio (95% confidence interval); * adjusted for smoking, drinking habits, body mass index, and medication; ** adjusted for baseline geriatric depressive symptoms score, job engagement, living alone, hobby or lesson, volunteering, dietary pattern, education level, and medication for lifestyle diseases.

**Table 4 nutrients-16-02179-t004:** Sensitivity analysis to determine the relationship between knee extensor muscle strength and depressive symptoms (GDS ≥ 6).

	Knee Extensor Muscle Strength (N)			*p* for Trend
	Lowest	Middle	Highest	
Crude adjusted	1.00 (reference)	0.57 (0.31, 1.05)	0.27 (0.13, 0.59)	<0.001
Age adjusted *	1.00 (reference)	0.59 (0.31, 1.12)	0.30 (0.13, 0.67)	0.002
Multivariable adjusted **	1.00 (reference)	0.75 (0.38, 1.45)	0.37 (0.16, 0.86)	0.022

Values are expressed as odds ratio (95% confidence interval); * adjusted for smoking, drinking habits, body mass index, and medication; ** adjusted for baseline geriatric depressive symptoms score, job engagement, living alone, hobby or lesson, volunteering, dietary pattern, education level, and medication for lifestyle diseases.

## Data Availability

The raw data supporting the conclusions of this article will be made available by the authors on request.

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
