# Peer review of "Knee Extensor Muscle Strength Associated with the Onset of Depression in Older Japanese Women: The Otassha Study"

_nutrients, 2024, doi:10.3390/nu16142179_

Round 1
Reviewer 1 Report
Comments and Suggestions for Authors
The current manuscript describes a two-year prospective cohort study that looked for the association between lower extremity muscle strength and the onset of depressive symptoms among elderly female individuals from Japan. The title and the abstract adequately reflect the study content!
The introduction section appears well-written. This section gradually introduces the topic, describes the problem statement and current state of affairs, and eventually flows to the inquiry question mentioned at the end of the section. This section does a good job of expanding on the complex relationships between physical fitness and mental health.- a relatively well-researched yet less understood association. The introduction section discusses the adverse effects of geriatric depression and adequately discusses the common precipitating factors. However, the discussion on lower extremity muscle strength needs more explanation and backing from peer-reviewed studies. Please elaborate on why lower-extremity muscle strength (knee extensor muscle strength) is considered as a surrogate marker for muscle strength and physical activity. The inquiry section adequately describes the purpose and the objectives of the current study in terms of whether there is a relationship between the two factors.
The methods section adequately describes the study plan. This study uses previously collected data from a large-scale observational study called as the Otassha Study, conducted in Tokyo, Japan. Given the fact that the current study uses already collected data from a previous large-scale study, it should rather be called a retrospective cohort study. Please explain as to why this study is a prospective cohort study. The instruments used for data collection have been adequately described. The statistical analysis should also include the software that was/were used for the data analyses.
The result section describes the study findings and concurs with the information presented in the tables and figures. The tables and figures appear clear and legible! This study has 1845 participants which is a significantly large sample, thus improving the external validity and generalizability! Figure -1 describes the process of how the patients were selected. Baseline characteristics have been described adequately in Table 1. Both crude and adjusted regression analyses have been reported in the manuscript as well as the tables.
The discussion section adequately summarizes the main findings of the study. It has summarized all major findings in the study and elaborated, compared, and contrasted them with findings in similar studies or research settings.
The conclusion section adequately summarizes the study findings. However, the implications of these findings for future research on the prevention and management of geriatric depression in this population have not been adequately discussed and need to be elaborated.
In general, this study is well-written except for the small changes recommended in the respective sections.
Author Response
Please elaborate on why lower-extremity muscle strength (knee extensor muscle strength) is considered as a surrogate marker for muscle strength and physical activity.
Your observation pertains to why lower extremity muscle strength serves as a surrogate marker for physical activity, a point we have not explicitly stated in the text. Our stance is that lower extremity muscle strength reflects a part of physical activity (https://pubmed.ncbi.nlm.nih.gov/33607291/). Furthermore, while numerous studies have utilized grip strength as a measure of exposure, we question its sufficiency. By quantifying muscle strengths beyond grip strength, we aimed to enhance and deliver insight into our understanding of the mechanisms that link muscle strength to mental health.
Additionally, we have presented and explained literature in our manuscript showing that the relationship between muscle strength and depression onset, is stronger for lower extremity muscle strength than for grip strength. This suggests that the association between lower extremity muscle strength and geriatric depressive symptoms observed in our study may indicate that the measured muscle strength partially reflects physical activity.
Please explain as to why this study is a prospective cohort study.
I appreciate your insights regarding the classification of our study. The Otassha study is indeed designed as a prospective cohort study, following participants over time to assess the association between baseline exposure factors and subsequent health outcomes. This design aligns with methodologies employed in established studies such as the UK Biobank. We have previously reported findings in this field using a similar prospective cohort approach (https://pubmed.ncbi.nlm.nih.gov/38704920/, https://pubmed.ncbi.nlm.nih.gov/37579925/, https://pubmed.ncbi.nlm.nih.gov/32894245/). Based on this consistency in study design and methodology, it is appropriate to describe the Otassha study as a prospective cohort study.
The statistical analysis should also include the software that was/were used for the data analyses.
We have inserted "Statistical analyses were conducted using R version 4.2.2." in the text at the point you specified.
The conclusion section adequately summarizes the study findings. However, the implications of these findings for future research on the prevention and management of geriatric depression in this population have not been adequately discussed and need to be elaborated.
As you pointed out, the conclusion section was missing some insights that could be gleaned from this study. An " Our study suggests that interventions aimed at enhancing muscle strength may help prevent the onset of depressive symptoms in older adults. Based on our findings, the focus should be on developing exercise programs specifically targeted at increasing muscle strength. Such programs could potentially be incorporated into routine geriatric care to mitigate the risk of depression. Additionally, this study will likely play a crucial role as a new insight into further elucidating the relationship between skeletal muscle and mental health." has been inserted to address your point.
Reviewer 2 Report
Comments and Suggestions for Authors
An issue affecting both physical and mental health is the correlation between depression in elderly women and the strength of their lower limb muscles. Studies have demonstrated a strong correlation between an older adult's total mobility, independence, and quality of life and physical strength, especially in the lower body. The authors described a prospective cohort study to evaluate the connection between older Japanese women's knee extensor muscle strength and the onset of depression.
The role of antidepressants in this unique demographic should be taken into account by authors (10.1186/s12877-020-01730-5) in the introduction. Findings indicates that having strong muscles may help one achieve a healthy lifespan and is important in preventing depression in later life confirming previuous evidence and suggesting muscle strenght as a risk factor for depression.
The documented and accurate indicator of total muscular strength, especially in older adults, is handgrip strength. This short, easy, and non-invasive evaluation can provide important details about a person's overall health and functional state. Morover, the implementation with lower limbs could be appropriated a generalization from a very local observation is limitated.
Furthermore, taking into account this topic and journal, authors should consider the impact of celiac disease, which can have a significant impact on muscle strength due to nutritional deficiencies, chronic inflammation, malabsorption, that in this frail population, should be assessed using non-invasive techniques (10.1097/MCG.0b013e318159c654). For those older patients, maintaining muscle strength entails maintaining a strict gluten-free diet, nutritional supplements, frequent monitoring, and adequate physical activity. When all of these aspects are taken care of those older adults can have improvements in their overall quality of life, muscle strength, and depression (10.3389/fnut.2022.838364).
Author Response
The role of antidepressants in this unique demographic should be taken into account by authors (10.1186/s12877-020-01730-5) in the introduction.
I agree that your perspective is significant. I have inserted an "Additionally, considering that antidepressant prescriptions account for less than 50% of prescriptions in late-life adults diagnosed with depression—who are at increased risk for polypharmacy—it is imperative to identify and study modifiable risk factors for depression [5]" in the introduction section.
The documented and accurate indicator of total muscular strength, especially in older adults, is handgrip strength. This short, easy, and non-invasive evaluation can provide important details about a person's overall health and functional state. Morover, the implementation with lower limbs could be appropriated a generalization from a very local observation is limitated.
Indeed, your insight is invaluable. The generalizability and simplicity of grip strength measurements offer distinct advantages that are not comparable with those of lower extremity muscle strength. However, age-related muscle weakness is often preceded by a decline in lower extremity muscle strength, which reflects a part of physical activity, rather than grip strength. Thus, assessing lower extremity muscle strength also has some advantages over grip strength. Additionally, this literature (https://pubmed.ncbi.nlm.nih.gov/33607291/) also cautions against using grip strength as the sole measure of muscle strength in geriatric assessments.
Nonetheless, the findings of this study are not intended to suggest that lower extremity muscle strength is a superior marker of depressive symptoms compared to grip strength. Instead, they underscore the importance of evaluating lower extremity muscle strength alongside grip strength. As these assessments target different muscle groups, they provide complementary insights, thereby enriching future research into the connections between muscle strength and mental health.
In light of the above, "Additionally, assessing lower extremity muscle strength is less convenient than measuring grip strength and lacks standardization. Therefore, further accumulation of knowledge in this field is essential." has been inserted into the limitations section.
Furthermore, taking into account this topic and journal, authors should consider the impact of celiac disease, which can have a significant impact on muscle strength due to nutritional deficiencies, chronic inflammation, malabsorption, that in this frail population, should be assessed using non-invasive techniques (10.1097/MCG.0b013e318159c654). For those older patients, maintaining muscle strength entails maintaining a strict gluten-free diet, nutritional supplements, frequent monitoring, and adequate physical activity. When all of these aspects are taken care of those older adults can have improvements in their overall quality of life, muscle strength, and depression (10.3389/fnut.2022.838364).
It is clear that the points you raised are indeed concerning. Thank you again for the references and for aiding our understanding of this area. Our cohort study did not assess a history of celiac disease. Considering the estimated prevalence of celiac disease in the Japanese population being about 0.05% (https://pubmed.ncbi.nlm.nih.gov/28389733/); the impact of the disease on our study results and conclusion must be negligible. Consequently, we have inserted an "It may also be necessary to consider including immune-mediated diseases, such as celiac disease, which manifests with symptoms of indigestion, in the investigation. Notably, multifaceted interventions, including a strict gluten-free diet and supplementation, have been reported to effectively alleviate depression in patients with celiac disease. However, given that the prevalence of celiac disease in Japan is approximately 0.05%, the results of this study are unlikely to be impacted by this disease. Further validation with a larger sample size is needed." in the Discussion-Limitations section to note this.
Reviewer 3 Report
Comments and Suggestions for Authors
Thank you for the opportunity to review this paper, which is a very well written article and represents an important contribution to the field. It presents the results of a prospective study performed on 768 women of old age, finding that a decline in muscle strength resulted associated in a linear way with the onset of depressive symptoms. Authors conclude that improving physical fitness in old ages can represents a modifiable lifestyle factor that can help to contrast the development of depression.
Introduction clearly proposes the background of the study. Materials and methods are appropriately described, but it is not explained how diet patterns presented in tables 1 and 2 have been defined. Discussion proposes some pathophysiological and social-life-related hypotheses to explain the results, which are consistent with the research in this field.
These few suggestions need addressing prior to publication.
Author Response
Introduction clearly proposes the background of the study. Materials and methods are appropriately described, but it is not explained how diet patterns presented in tables 1 and 2 have been defined.
A "The dietary pattern score was calculated based on the frequency of consumption of soy products, fruits, and green-yellow vegetables. Participants received 1 point for rare consumption, 2 points for consuming these items once or twice a week, 3 points for consumption every other day, and 4 points for almost daily consumption. The total possible score ranged from 3 to 12, with higher scores suggesting a dietary pattern more conducive to mitigating depression." has been inserted in the appropriate section due to the omission of an explanation for an important variable.
Round 2
Reviewer 2 Report
Comments and Suggestions for Authors
Authors addressed all comments
Author Response
Thank you for your comments and opinions on our manuscript.
Reviewer 3 Report
Comments and Suggestions for Authors
Authors have accurately answered the questions posed, and the revised version of the manuscript in its current version can be accepted for publication.
Author Response

(The authors gave the same response as above.)
